# A Review of Carbapenem Resistance in *Enterobacterales* and Its Detection Techniques

**DOI:** 10.3390/microorganisms11061491

**Published:** 2023-06-03

**Authors:** Oznur Caliskan-Aydogan, Evangelyn C. Alocilja

**Affiliations:** 1Department of Biosystems and Agricultural Engineering, Michigan State University, East Lansing, MI 48824, USA; oznurca@msu.edu; 2Global Alliance for Rapid Diagnostics, Michigan State University, East Lansing, MI 48824, USA

**Keywords:** antibiotic resistance, carbapenem resistance, carbapenem-resistant *Enterobacterales* (CRE), carbapenemases, detection technology, surveillance

## Abstract

Infectious disease outbreaks have caused thousands of deaths and hospitalizations, along with severe negative global economic impacts. Among these, infections caused by antimicrobial-resistant microorganisms are a major growing concern. The misuse and overuse of antimicrobials have resulted in the emergence of antimicrobial resistance (AMR) worldwide. Carbapenem-resistant *Enterobacterales* (CRE) are among the bacteria that need urgent attention globally. The emergence and spread of carbapenem-resistant bacteria are mainly due to the rapid dissemination of genes that encode carbapenemases through horizontal gene transfer (HGT). The rapid dissemination enables the development of host colonization and infection cases in humans who do not use the antibiotic (carbapenem) or those who are hospitalized but interacting with environments and hosts colonized with carbapenemase-producing (CP) bacteria. There are continuing efforts to characterize and differentiate carbapenem-resistant bacteria from susceptible bacteria to allow for the appropriate diagnosis, treatment, prevention, and control of infections. This review presents an overview of the factors that cause the emergence of AMR, particularly CRE, where they have been reported, and then, it outlines carbapenemases and how they are disseminated through humans, the environment, and food systems. Then, current and emerging techniques for the detection and surveillance of AMR, primarily CRE, and gaps in detection technologies are presented. This review can assist in developing prevention and control measures to minimize the spread of carbapenem resistance in the human ecosystem, including hospitals, food supply chains, and water treatment facilities. Furthermore, the development of rapid and affordable detection techniques is helpful in controlling the negative impact of infections caused by AMR/CRE. Since delays in diagnostics and appropriate antibiotic treatment for such infections lead to increased mortality rates and hospital costs, it is, therefore, imperative that rapid tests be a priority.

## 1. Introduction

Antimicrobial resistance (AMR) is acquired when microorganisms grow or survive in the presence of antimicrobials or drugs designed to kill them [1]. AMR threatens the effective prevention and treatment of a wide range of infections caused by pathogenic bacteria, viruses, parasites, and fungi. AMR has been a serious threat to public health since the beginning of the last decades [1,2,3,4,5].

AMR has a high potential to increase costs and destabilize the health infrastructure. A recent report by the Centers for Disease Control and Prevention (CDC) in 2019 stated that AMR kills at least 1.27 million people worldwide and is associated with approximately 5 million deaths [1,2]. In the United States (US), the CDC reported that AMR causes more than 2.8 million infections and 35,000 deaths annually [1], with a predicted annual cost of approximately USD 55 billion [1,6]. In Europe, AMR results in an estimated 25,000 deaths and a cost of EUR 1.5 billion in health expenditures each year [4]. In accordance with recent estimates, infections by antimicrobial-resistant microorganisms will annually result in 10 million deaths, along with USD 100 trillion in costs, by the year 2050 [7]. The problem of AMR is particularly urgent due to the high presence of unregulated antibiotics in the market [1,3,8]. The misuse and overuse of antibiotics enable the emergence and spread of resistance in bacteria, leading to more difficulty in controlling and treating such infections [5].

### 1.1. Development of Antibiotic Resistance and Their Mechanisms

The first antibiotic, penicillin, was found by Alexander Fleming in 1928. With its release into the market in 1941, penicillin-resistant bacteria (*Staphylococcus aureus*) increased in the following year [1,9]. There has been a continued discovery of new antibiotics coupled with the emergence of resistance, since bacteria find ways to survive and resist new antibiotics, resulting in less-effective drugs. Antibiotics have different strategies and mechanisms for bacterial death [1,9,10]. In general, bacteria have different mechanisms to become resistant to antibiotics, and they are listed as follows: (1) restriction on the access of the antibiotic by changing the entry pathways or limiting their number, (2) activation of efflux pumps in their cell walls to remove the antibiotics that enter the cell, (3) changing or destroying the antibiotic with enzymes, (4) alteration of the targets for the antibiotic by modification of intracellular enzymes so that they can no longer latch onto it, and (5) the development of a new cell process to bypass the effects of the antibiotic [3,4,11,12,13]. Bacteria can further form biofilms on surfaces by extracellular enzymes to prevent antibiotics from penetrating through the outer cell membrane, allowing the natural growth of the bacteria [14]. Thus, once microorganisms are exposed to antibiotics, they adapt and grow in the presence of antibiotics, similar to their adaptation to a new environment. They use their genetic mechanisms to increase their adaption [4,8].

To understand the problem of antibiotic resistance, it is helpful to discuss how antibiotic resistance is developed in a bacterial population, which is the main focus of infections caused by antibiotic-resistant bacteria in clinical settings. Acquired resistance is developed in a bacterial population that is originally susceptible to antibiotics either by genetic mutations and the acquisition of resistant genes or through horizontal gene transfer (HGT) [10,13,15,16,17]. In the former route, a subset of bacterial cells from susceptible populations, in the presence of antibiotics, can develop mutations, allowing survival. When the mutation emerges, resistant subpopulations become dominant [13,14,15] as a result of bacterial multiplication (vertical evolution), passing the gene on to their generations [14,15]. A few studies on the enrichment of resistance genes and mutation selection showed that the time scale of acquiring antibiotic resistance was extended up to 24 transfers [18], 20 days [19], and 1000 generations, which take 10–15 days [20].

Genes that encode resistance are commonly transferable; acquiring genes by HGT is a significant driver of bacterial evolution and is primarily responsible for the spread of antibiotic resistance [13,14,15,21]. The transmission of genetic materials between microorganisms occurs through three main routes: (1) transformation, incorporation of free DNA from the environment into the chromosome; (2) transduction, phage-mediated transfer of DNA from infected cells through virus particles; and (3) conjugation, transfer of plasmids from one bacterium to another. Conjugation is the most efficient and common way to share genetic information, always leaving behind a copy of the resistant gene, resulting in the emergence and spread of AMR [11,13,14,15]. Resistant genes can also be acquired by transposons or integrons linked with mobile genetic elements (MGE). Transposons, specialized fragments of DNA, carry several resistant genes but cannot replicate by themselves. They can move within the genome, facilitating resistant gene migration from the chromosome to the plasmid [14,15]. Integrons can also encode several resistant genes but are not movable. Thus, encoding mechanisms are based on the capture of resistant genes and the excision of the genes within and from the integrons. This is one of the efficient mechanisms of the accumulation of resistant genes. Integrons also provide a mechanism for adding new genes into the bacterial chromosome and are mostly carried in plasmids, increasing the horizontal mobility of the antibiotic-resistant genes [13,15,17].

### 1.2. Factors Converging Emergence and Transmission of Antibiotic Resistance

The development of antibiotic resistance worldwide is increasing due to the misuse and excessive use of antibiotics and antifungals, global trade networks, medical tourism, poor sanitation conditions, improper waste management systems, and urbanization [21,22,23,24,25,26]. Remarkably, the overuse and misuse of antibiotics in healthcare, veterinary medicine, agriculture, and aquaculture and their release to the environment contribute to the emergence and spread of AMR [21,23,24,25,26]. Significant sources of antimicrobial-resistant bacteria include healthcare settings and the environment. AMR can be transmitted through contact with people, animals, and contaminated water or foods [27]. In addition, intestinal commensal bacteria have been reported as a significant reservoir of antimicrobial-resistant bacteria and genes (ARGs). Due to HGT and the prior use of antibiotics, the commensal flora of humans and animals can acquire ARGs; the fecal carriage of resistant bacteria and ARGs leads to their emergence and spreads in the community, environment, animal, and foods [27,28]. For example, the surveillance of human fecal carriage has shown a significant increase in intestinal ARG carriage worldwide [27]. In another example, animal guts can contaminate its products during animal slaughtering and food processing. The handling and consumption of contaminated food or contact with animals or their surroundings (fertilizer) cause the spread of AMR in the community and environment (soil and water), along with fruits, vegetables, etc. [1]. Thus, wastewater from human activities, healthcare services, and general population-collected wastewater treatment plants (WWTP) are sources of antimicrobials, commensal and pathogen bacteria, antibiotic-resistant bacteria, and ARGs [26]. Due to inefficient, inappropriate, or missing regulatory status and practices on WWTP systems, contaminated urban wastewater, sewage sludge, manure, sediment, and reclaimed water result in their accumulation and spread in the environment and community [11,26]. As seen in Figure 1, everything is connected in a complex web; the health of people is connected to the health of animals and the environment. Hence, communities, healthcare facilities, environments, food, farms, and animals are all impacted, affecting progress in healthcare and life expectancy [1,26].

AMR causes threats to anyone regardless of age, to immunocompromised people, and to people with chronic illnesses [1,29]. AMR also puts at risk those who receive modern healthcare advances, such as joint replacements, organ transplants, cancer therapy, etc. These procedures have a risk of infection, and effective antibiotics may not be available [1,30]. AMR is a global crisis, and new forms of resistance emerge and rapidly spread across countries and continents through people, goods, and animals. One billion people travel through international borders every year, and a global effort is necessary to slow the emergence and spread of AMR [1,5].

## 2. Urgent Threat of Infections by Antimicrobial Resistant Bacteria: Carbapenem-Resistant Bacteria

The World Health Organization (WHO) and CDC reported the current and future threat of infections by antimicrobial-resistant microorganisms with a high level of concern [1,31]. Carbapenem-resistant *Acinetobacter baumannii* (CRA), carbapenem-resistant *Pseudomonas aeruginosa* (CRP), and carbapenem-resistant *Enterobacterales* (CRE) have been listed as critical priority pathogens by the WHO [31]. In addition, CRE and CRA have been reported as the most urgent threats by the CDC since 2019 [1]. Particularly, CRE results in 1100 deaths and 13,100 infections in the USA [1], with a high fraction of these infections potentially resulting in death due to limited antibiotic therapies [1,22,30].

Carbapenems, a broad-spectrum β-lactam antibiotic, are structurally related to penicillin [32]. Carbapenems have a carbon instead of a sulfone at the fourth position of the β-lactam ring, differing from other β-lactams. The unique structure plays a major role in their stability against β-lactamases [33]. Carbapenems are not easily diffusible through the cell wall, but they enter the bacteria through outer membrane proteins (porins). Then, carbapenems degrade the cell wall at the penicillin-binding proteins (PBPs) via the β-lactam ring. The mode of action weakens the glycan backbone in the cell wall due to autolysis, and the cell is destroyed because of osmotic pressure [32,33,34].

Carbapenems have been used as last-line agents against Gram-negative, Gram-positive, and anaerobic bacteria [33]. The last-resort antibiotics were approved for clinical use in humans and released into the market in 1985 [1,35,36]. Carbapenems may occasionally be used for pets under certain conditions, according to the Animal Medicinal Drug Use Clarification Act (AMDUCA) [37,38]. Among carbapenems, ertapenem and panipenem have limited use against non-fermentative Gram-negative bacteria but are appropriate for community-acquired infections. Carbapenems, including imipenem, meropenem, doripenem, and biapenem, have been widely used in hospital-acquired infections. These carbapenems are typically reserved for use in patients infected with multi-drug resistant (MDR) bacteria, including extended-spectrum β-lactamase (ESBL)-producing and ampicillinase C (AmpC)-producing bacterial infections [22,23,34], such as complicated intraabdominal and urinary infections, bloodstream and skin infections, community-acquired and nosocomial pneumonia, meningitis, and febrile pneumonia [35,39,40].

Carbapenem-resistant bacteria were first described in 1996 with the identification of carbapenemase-producing *Klebsiella pneumoniae* [11,23]. In the last decade, the emergence and spread of carbapenem-resistant bacteria have globally increased. For example, many infections caused by CRE are mostly seen in patients in hospitals, long-term care facilities, and long-term acute care hospitals [41,42,43]. Such infections are high risk for patients using ventilators, urinary catheters, intravenous catheters, and long-term antibiotic treatment and for immunocompromised patients [41]. A significant fraction of these infections result in death due to limited treatment options [1,22,30,44]. Specifically, bloodstream infection by CRE causes a high mortality rate in pediatric populations [40,45]. The characteristics, mechanisms, and outcomes of carbapenem-resistant bacteria are thus crucial to prevent and manage such infections [46].

CRE are *Enterobacterales* resistant to at least one of the carbapenem antibiotics based on their antibiotic susceptibility profile (phenotypic definition) [41]. There are different mechanisms (e.g., genotypic); carbapenem resistance mainly develops when bacteria (1) acquire structural changes in penicillin-binding proteins (PBPs), (2) show a decrease or loss of specific outer membrane porins that filter carbapenems from reaching their site of action, (3) activate the efflux pumps to remove the antibiotics and regulate the intramembrane environment, and (4) acquire β-lactamases and carbapenemases to degrade or hydrolyze carbapenems and other β-lactam antibiotics (e.g., penicillins and cephalosporins) [32,33,34,41]. In addition, carbapenem resistance can be acquired by a combination of CTX-M (activity against cefotaxime) and AmpC enzymes, allowing low-level carbapenem resistance. Further, the combination of the β-lactamase expression and porin gene mutations is associated with high-level carbapenem resistance, attenuating therapy responses [47].

Overall, CRE can become resistant through chromosomal mutations in the porin gene (non-carbapenemase-producing CRE) and/or the production of carbapenem hydrolyzing-enzymes (carbapenemase-producing (CP) CRE) [41]. The presence or expression of the gene coding carbapenemase is usually sufficient for carbapenem resistance, covering 30% of CRE. Thus, CP-CRE is a subset of all CRE [22,41]. These genes are often on mobile genetic elements, leading to their rapid spread and resulting in infections and colonization [1,14,41,48]. Many CRE-colonized individuals do not develop infections; however, they can still spread the bacteria [41]. Similarly, the transfer of genetic elements can occur in the food chain and the environment [1,14,41]. Therefore, routine tests for these carbapenemases through the Antibiotic Resistance Laboratory Network and CDC laboratories are conducted to prevent and control their emergence and spread [41].

### 2.1. Carbapenemases

A large variety of carbapenemases have been classified into three groups: Ambler Classes A, B, and D β-lactamases, based on hydrolytic and inhibitor profiles using active catalytic substrates of serine or zinc [13,23,32,34,49]. The characteristics of the three most common classes of carbapenemases are detailed and listed in Table 1 [23,49].

Class A enzymes, serine β-lactamases, hydrolyze a broad variety of β-lactam antibiotics, including carbapenems, cephalosporins, penicillin, and aztreonam [49]. These enzymes were identified as chromosomally encoded and plasmid-encoded types [49]. Some of the chromosomally encoded genes are NmcA (not metalloenzyme carbapenemase A), SME (*Serratia marcescencens* enzyme), IMI-1 (imipenem hydrolyzing β-lactamase), and SFC-1 (*Serratia fonticola* carbapenemase-1). The plasmid-encoded genes are KPC (*Klebsiella pneumoniae* carbapenemase), IMI (Imipenem-hydrolyzing beta-lactamase), and GES (Guiana extended spectrum) [23,32]. Among these, the KPC type is the most prevalent enzyme and causes outbreaks in many Asian, African, North American, and European countries [23,32]. KPC gene is mainly located within a 10-kb length, mobile transposon Tn4401, frequently established on conjugative plasmids. The link of *bla_KPC_* with plasmids and transposons assists in intraspecies gene transfer and the dissemination of the gene [50]. Several KPC variants have rapidly increased, and 84 KPC alleles have been recorded in the GenBank database [51]. Of these, KPC-2 and -3 are the most common enzymes worldwide, and 22 KPC variants have also conferred ESBL-, CTX-M-, or ceftazidime-avibactam (CZA)-resistance in their gene position. For example, the KPC-2 gene was carried on the NTE_KPC_-Ib transposon on plasmids with a 15-bp insertion, which also harbored the resistance gene, CZA resistance [47,51]. Overall, KPC types are mostly found in *Klebsiella pneumoniae*, *Klebsiella oxytoca*, *E. coli*, and *Serratia marcescens*, as well as in *Enterobacter*, *Salmonella*, and *Proteus* species [13,23,32]. Their rapid spread and diverse variants severely threaten human health and impact therapeutic efficacy [13,32,51].

Class B enzymes are known as Metallo-β-lactamases (MBL) since they utilize metal ions (usually Zinc) as a cofactor to attack the enzyme’s active site (β-lactam ring). There are 10 types of MBLs; the most important ones include New Delhi Metallo-beta-lactamase (NDM), Verona Integron-Encoded Metallo-beta-lactamase (VIM), and Imipenemase (IMP) [23,32,41,52]. They hydrolyze all current β-lactam antibiotics, except for monobactams (e.g., aztreonam) [53]. IMP was first reported in Japan in *S. marcescens* in the early 1990s [13], and over 85 sequence variants have been described [53]. IMP variants are found in *Acinetobacter* and *Pseudomonas* species, as well as in the *Enterobacteriaceae* family [13,32]. VIM was then identified in *P. auregionasa* in Verona, Italy, in late 1997, and over 69 variants have been described [53]. VIM variants are mostly found in *Pseudomonas*, *Acinetobacter*, and *Enterobacteriaceae* species, which are globally distributed [13,32,53]. Recently, NDM was the most prevalent MBL, first identified in *Klebsiellea pneumoniae* and *E. coli* isolated from a patient who traveled from India to Sweden in 2008 [13,53]. There have been 29 NDM variants described, and NDM-1 is the most prevalent type. NDM variants are generally dominant in *Klebsiella pneumoniae*, *E. coli*, *Acinetobacter baumannii*, and *Pseudomonas aeruginosa* [32,52,53].

Class B enzymes are usually found in plasmid vectors or other mobile genetic elements [49]. For instance, IMP and VIM are mostly integron-associated; they are encoded by gene cassettes within class 1 or 3 integrons that may be embedded in transposons, allowing insertion into the bacterial plasmids [53]. NDM is not integron-associated; it has been observed in plasmids rapidly disseminated worldwide [52,53]. Additionally, NDM-producing bacteria can have both NDM-1 and a type IV secretion system (T4SS) gene cluster in plasmids, showing high virulence [52]. Further, NDM-producing bacteria may harbor other carbapenemases in plasmids (e.g., KPC, VIM, and OXA types) and ESBLs [13,41,47,52]. Thus, the emergence of NDM-producing bacteria with increasing variants is a significant threat to public health.

**Table 1 microorganisms-11-01491-t001:** The most common carbapenemases in bacteria with their gene location [23,49,53].

Ambler Class	Representative Gene	No of Variants	Gene Location	Bacterial Origins
A	KPC (Klebsiella pneumoniae carbapenemase)	>84	Plasmid	*K. pneumoniae*
	GES (Guiana extended spectrum)	>27	Plasmid	*P. aeruginosa*
	IMI (Imipenem-hydrolysing beta-lactamase)	>9	Chromosome	*E. cloacae*
	SME (*Serratia marcescencens* enzyme)	>5	Chromosome	*S. marcescencens*
	SFC (*Serratia fonticola* carbapenemase-1)	>1	Chromosome	*S. fonticola*
	NMC-A (not metalloenzyme carbapenemase A)	>1	Chromosome	*E. cloacae*
B	NDM (New Delhi metallo-lactamase)	>29	Plasmid	*K. pneumoniae*
	VIM (Verona integron-encoded metallo-lactamase)	>69	Plasmid	*P. aeruginosa*
	IMP (Imipenemase),	>85	Plasmid	*S. marcescencens*
	GIM (German imipenemase)	>2	Plasmid	*P. aeruginosa*
	SIM (Seoul imipenemase)	>1	Plasmid	*P. aeruginosa*
D	OXA (Oxacillin-hydrolyzing carbapenemase)	>40	Plasmid	*K. pneumoniae*

Class D enzymes, serine β-lactamases, are oxacillinase or oxacillin-hydrolyzing enzymes (OXA), comprising over 200 enzymes. OXA rapidly mutates and expands its spectrum activity; the most prevalent carbapenem-hydrolyzing enzymes are OXA-48 and OXA-181 in over 40 carbapenemase variants [36]. OXA-48 was first identified in *Klebsiella pneumoniae* in Turkey in 2001 [54,55]. Plasmids are the primary genetic elements for the transmission and propagation of the genes; the most frequent hosts for OXA-48 are self-conjugative 60- to 70-kb plasmids [55]. Currently, OXA-48 and OXA-101 variants are mostly dominant in *Klebsiella pneumoniae* in Turkey, the Middle East, North Africa, and Europe [13,32,36,55]. However, it should be noted that OXA-producing bacteria often have low-level resistance due to weak expression, which is risky for false positive detection and suitable treatment options [55].

The genes coding carbapenemase in β-lactamase (*bla*) are defined as *bla_KPC_*, *bla_NDM_*, *bla_OXA-48_*, *bla_VIM_*, and *bla_IMP_* [13,34,56]. These genes are found in many bacteria, such as *E. coli*, *K. pneumoniae*, *Salmonella*, *Acinetobacter*, and *Pseudomonas*. These bacteria are isolated not only from humans but also animals, food supplies, and water sources worldwide [23,48,57], detailed in the next section.

### 2.2. Dissemination of the Carbapenemases in Humans, Animals, Foods, and Environment

Several studies have shown that healthcare settings can lead to the spread of CP pathogens in humans [23,29,58]. Frequent hospital visits and long-term stays in healthcare facilities represent a high risk of colonization and infection development with CP bacteria, particularly with CP-CRE [23,29]. For instance, KPC-producing *K. pneumoniae* caused hospital outbreaks in many European countries such as Greece, Italy, Spain, France, and Germany [59,60,61]; NDM and KPC-producing *K. pneumoniae* were identified in transplanted patients in Brazil [62]; CP-CRE were found to spread in hospital and community settings in Africa [63,64,65] and Asia [58,66,67]. Another factor of CP-CRE spread is international travel and medical tourism [23]. For instance, KPC-producing *K. pneumoniae* and *E. cloacae* were isolated from patients in New York who had recently traveled from France and Greece [68,69]. In another example, NDM-producing *K. pneumoniae* and *E. coli* were isolated from Sweden and UK patients who recently traveled to India [70,71].

Among CP-CRE, *E. coli* and *K. pneumoniae* have been disseminated globally at an alarming rate in the medical community as critical human pathogens [72,73]. For instance, KPC-producing *K. pneumoniae* has been found in more than 100 different sequence types (STs). Particularly, *K. pneumoniae* ST258 is predominant and primarily associated with KPC-2 and KPC-3 production. ST258 comprises two distinct lineages, clades I and II, and ST258 is a hybrid clonal complex created by a large recombination event between ST11 and ST442 [73]. Further, ST11, ST340, and ST512 are single-locus variants of ST258 and harbor carbapenemases. ST11 is closely related to ST258, which is associated with KPC, NDM, VIM, IMP, and OXA-48 production [73].

Further, carbapenem resistance in pathogenic *E. coli* is a major concern because of limited therapy. For instance, *E. coli* ST131, causing severe urinary infections, has been linked to the rapid global increase in AMR among *E. coli* strains [72]. Further, FimH30 lineage and virotype C are the common lineage among ST131, contributing to the spread of ST131 associated with carbapenemases. ST131 is most likely responsible for the global distribution of *E.coli* with KPC, NDM, and OXA-48 production [72]. These sequence types of *E. coli* and *K. pneumoniae* pose a major threat to public health because of their worldwide distribution [72,73].

Additionally, hospitals or health-care settings are a reservoir for CP bacteria. Carbapenem residues in human excreta can get into hospital sewage. Due to the selection of a low concentration of antibiotics, bacteria in hospital effluent can become resistant to carbapenems [23]. Hospital sewage may act as a reservoir for resistance genes, where bacteria likely acquire resistance through HGT [23,29]. Likewise, antibiotic residues and resistant genes released into municipal wastewater could contribute to the selection of CRE and their dissemination to ground and surface water, spreading them to the environment [23]. For example, CP *E. coli*, *E. cloacae*, *K. pneumoniae*, and *Citrobacter freundii* were found in the river and hospital sewage in Portugal [74], China [75], Vietnam [76], and Australia [77,78]. VIM- and KPC-producing *E. coli* were found in seven waste water treatment plants in the USA [79]; OXA-48 carrying CRE in tap water was found in six states in the USA [78]. In addition, KPC-producing *Salmonella* was found in human feces, hospital sewage, and effluent in the USA and Brazil [80,81].

Another possible way of CP bacteria transmission to animals and farms is through direct contact with colonized hosts (human and animal) and a contaminated environment (surface water, ground water, soil) [23]. CP bacteria (*E. coli*, *K. pneumoniae*, *Salmonella*, *Acinetobacter*, *Pseudomonas*) have been detected in farm animals, poultry, fish, mollusks, and wild birds and animals [23,48,57,82,83,84,85,86,87,88]. The transmission of CP bacteria also alerts food safety, particularly CRE in the food-chain. For instance, CP bacteria were isolated in meat (beef, chicken, pork), seafood (clam, fish, prawn), and vegetables (lettuce, spinach, Chinese cabbage, roselle) [42,89,90,91,92,93,94]. These studies showed major carbapenemases (NDM, VIM, and KPC) present in foods. The presence of CP bacteria in the food chain mainly contributes to their spread worldwide due to the global food trade, posing a risk to human health [93].

Various environmental, microbiological, and clinical investigations have shown that CP-CRE can widely spread in the community, animal and agricultural products, and the environment [23,38,83,95,96,97]. For the early detection and optimal management of the spread and emergence of CRE, some recommendations include (1) the necessity of screening and rapid diagnostic tools for patients who may have visited countries or hospitals with frequent infection by CRE, (2) specific policies and prioritizing funding for the control and management of infections by CRE, (3) clear strategies indicating the use of carbapenems, and (4) international co-operation to reduce the global spread of CRE [98]. As infections caused by particularly CRE are a global concern, the rapid detection of the causative bacteria is of utmost importance [38,46,98].

## 3. Current and Emerging Detection Techniques of CRE

Diagnostic tests assist in screening or monitoring specific infections or conditions to control and prevent CRE spread in the community. However, diagnostic AST protocols usually start with identifying the bacteria species in selective media, followed by growth in the presence of antibiotics (carbapenem) for determining their antibiotic-resistant profile [1,99]. However, each hour of delay in obtaining a correct diagnosis and appropriate antibiotic treatment of infections by CRE increases the mortality rate by approximately 8% [100]. For instance, delayed diagnosis and treatment in CP-CRE raise the mortality risk from 0.9% to 3.7%, hospital cost from ~USD 10,000 to ~USD 25,000, and hospital stay from 5.1 days to 8.5 days [43,101]. Thus, rapid and accurate detection is a significant step in controlling and preventing such microbial infections. Several culture-based, rapid phenotypic, genotypic methods, and biosensors have been developed to detect carbapenem resistance, including carbapenem-hydrolyzing enzymes, detailed in this section with advantages and limitations.

### 3.1. Culture-Based Methods

**The antimicrobial susceptibility testing (AST)** is widely used in clinical and public health laboratories to assess the antimicrobial resistance profiles of target microorganisms. The standard culture-based AST methods include broth and agar dilution tests, disk diffusion, and E-tests. These tests, approved by the Food and Drug Administration (FDA), involve the isolation of pure cultures of the potential pathogens, followed by testing these bacteria on media with minimum inhibitory concentration (MIC) levels [99,102,103,104]. Specifically, disk diffusion is a gold standard for AST; bacteria are inoculated on agar plates with a single antibiotic disk and then incubated to determine the resistant profile. Among carbapenems, imipenem, meropenem, and ertapenem have been commonly used for the early detection of carbapenem resistance; ertapenem has been described as the most sensitive indicator [105]. To determine the susceptible, intermediate, and resistant profile of tested bacteria, the most widely used standard interpretation of AST and breakpoints are recommended by the Clinical and Laboratory Standards Institute (CLSI) and European Committee for Antimicrobial Susceptibility Testing (EUCAST) [106].

**AST disk diffusion and E-test combination with specific inhibitors** have been used to differentiate the carbapenem-hydrolyzing enzymes from two main types, KPC and MBLs [107]. Examples of this are the addition of chelating agents, such as EDTA, in the broth microdilution and E-test aids in confirming the presence of MBLs with binding zinc ions and inhibiting MBL activities [49,108]. The sensitivity and specificity of this test are reported to be >82% and >97%, respectively [107]. Similarly, phenyl-boronic acid (PBA) is incorporated into the E-test for KPC identification by the inhibition of KPC activity. This test’s sensitivity and specificity are reported as 92% and 100%, respectively [107]. Multidisc diffusion tests with inhibitors of specific enzyme types, including clavulanate for ESBL and cloxacillin for AmpC, are also used to differentiate enzymes [32,107,108].

**The modified Hodge test (MHT)** was developed to identify the presence of carbapenemases [109,110]. The MHT was first introduced in 2010 for detecting carbapenemase genes and was widely used because of its ability to detect KPC producers. In this method, the suspected bacteria are inoculated by swabbing a straight line from the edge of the meropenem disk on Mueller Hilton Agar (MHA) that is pre-inoculated with susceptible *E. coli*. The plates are incubated overnight, and the cloverleaf-like zone is observed for CP isolates. The use of this method was recommended by CLSI in 2009. It has good sensitivity for other carbapenemases (VIM, IMP, and OXA-48), although its performance in detecting NDM enzymes was found to be lower [108,110]. Overall, its sensitivity and specificity were found to be 69% [110] and 93–98% [107], respectively.

**The carbapenemase inactivation method (CIM)** has been recently introduced by CLSI (2016) with higher accuracy and accessibility [107,108,111]. This method is initiated by a suspension of bacteria in a broth and incubation with a meropenem disk (2–4 h); if the isolate produces the enzymes, the meropenem in the disk is degraded. The disk in the broth can then be placed on MHA streaked with susceptible *E. coli* and incubated, which detects carbapenemase activity with no zone or a narrow zone diameter of <19 nm [107,108,111]. This method showed high concordance with results obtained by a PCR test, which is used in many clinical and public health laboratories [107,108]. The sensitivity and specificity of the CIM method were over 95% [107,111].

**Specific media** have also been designed for CP strain screening [32,112,113]. For example, Chromogenic Media and Brilliance CRE Agar are used for the initial detection of CRE strains in colonized and infected patients, with 76.5% sensitivity. CHROM agar KPC is used to screen for KPC and VIM-producing *Enterobacteriaceae*, but it can detect high-level resistance with 43% sensitivity. SUPERCARBA medium is mainly used for KPC and OXA-48 producers and is applicable to detect low-level resistance with higher sensitivity (96.5%) [112,113,114]. ID Carba and Colorex KPC media were designed for CP Enterobacteriaceae [112]. All these selective media are directly applicable to patient samples; however, they have lower specificity (>50%) depending on the enzyme type [112,114,115].

The mentioned culture-based methods are cost-effective and widely applicable. Among these methods, the CIM has a higher sensitivity and specificity in identifying and typing carbapenemases. However, they are labor-intensive and require time-consuming steps to isolate pure cultures, taking days to weeks to determine the resistance profile of the suspected bacteria [99,102,103].

### 3.2. Rapid Phenotypic Methods

**Automated AST systems, which are rapid culture-based methods**, have been developed to shorten the required time to detect antimicrobial resistance [103,105]. For example, FDA-approved commercial automated instruments are MicroScanWalkAway and Vitek-1/Vitek-2, which measure bacterial growth in the presence of antibiotics by recording bacterial turbidity using a photometer [32,103]. Further, BD Phoenix measures bacterial growth in the presence of antibiotics by recording bacterial turbidity and colorimetric changes. Sensititrere records bacterial growth with antibiotics by measuring fluorescence [103]. Besides imaging-based technologies, automated microscopes, such as multiplexed automated digital microscopy (MADM) that is FDA approved, single cell-morphological analysis (SCMA), oCelloscope, Fluorescence microscopy, and cell lysis-based methods are also used. These automated microscopes measure the phenotypic response, changes in bacterial growth rate, and the cellular morphology and structure profile of bacteria in the presence of antibiotics [32,103,104].

**Optical techniques** have also been developed, which measure the physical and biochemical profile of bacterial cells [116]. For example, forward laser light scatters (FLLS) and rapid electro-optical technology have been used to measure bacterial numbers by optical density and to estimate cell density and size using light scattering of the cell particles in a liquid [99,103,117]. Another optical technique, flow cytometry (FC), is used for cell counting and the detection of a biomarker using changes in morpho-functional and physiological characteristics of cells [103,118,119]. Additionally, Raman spectroscopic analysis has been recently used to measure and compare the spectra of bacteria in the presence of antibiotics to distinguish resistant strains [99,104,120]. Further, several miniaturized lab-on-a-chip systems have also been fabricated using microfluidic techniques, substituting agar to measure the growth of pure bacteria in the presence of antibiotics for rapid testing [99,103]. An ultraviolet (UV) spectrophotometric method was developed to measure the carbapenem imipenem hydrolysis activity of CP bacteria [121]. Lastly, bioluminescence-based detection assays (BCDA) have also been developed for CP bacteria based on adenosine triphosphate (ATP) level differences in culture media. Such assays are rapid (<2.5 h) and accurate, with higher specificity and sensitivity [122]. However, the applicability of this technique in matrices is low due to reduced sensitivity [108,121,122,123].

**Matrix-assisted laser desorption/ionization time-of-flight mass spectrometry (MALDI-TOF MS)** in optical techniques has recently become popular for identifying pathogens and resistant bacteria due to its distinct fingerprint spectra [104,118]. To determine the antimicrobial resistant profile in pathogens, MALDI-TOF MS identifies (1) the antimicrobial-resistant clonal group (e.g., carbapenem-resistant *E. coli*), the modified antimicrobial drug (e.g., carbapenemase activity), the modified antimicrobial target (e.g., lipid A modification), the direct detection of the AMR determinant (e.g., KPC-2 β-lactamases), and biomarkers co-expressed with the AMR determinants (e.g., *bla*_KPC_ carrying plasmid) [124]. This technique identifies specific resistant profiles (e.g., KPC and MBLs) of bacteria at the species and genus level from single isolated colonies within 1–4 h, with 72.5–100% sensitivity and 98–100% specificity; however, it has issues regarding OXA-48 identification [40,123,124]. Further, the combination of automation and the implementation of a user-friendly interface recently made MALDI-TOF MS popular in clinical laboratories [123,125,126,127]. 

**Colorimetric assays** have also been developed as rapid, simple, and cost-effective phenotypic methods for detecting CP bacteria based on their carbapenemase hydrolytic activity [60,107,108]. The Carba NP test (2 h) measures the hydrolysis of imipenem, leading to changes in pH and resulting in a color change from red to yellow/orange. The sensitivity of this test was found to be 73–100% for most carbapenemases, but it performed poorly in the detection of the OXA-48 enzyme [60,107,108]. The Carba NP test has been recommended for use as a first-line test for screening carbapenemase activity by the CLSI in the US. The RAPIDEC Carba NP test (first commercial test), β-CARBA test, Rapid CARB screen, Rapid Carb Blue kit, and Neo-CARB kit have been used to detect CP bacteria within 2 h, with varying sensitivity (>70%) and specificity (>89%) from the pure culture [108,123,128]. However, these assays require pure cultures and are dependent on the growth rate of the bacteria [123].

Many of these rapid phenotypic techniques and automated systems still require pure cultures; thus, sample preparation and pre-treatment steps require several hours to days [116,123]. Last but not least, these techniques require costly equipment, complex data analysis, and skilled personnel, which limits their applicability in low-resource setting laboratories [99,103,104,118].

### 3.3. Genotypic Methods

Molecular AST methods are effective techniques to detect specific resistant genes in a short time from matrices without the need for a tedious bacterial culture and a long incubation time [116,123]. Among these, PCR-based methods, DNA microarray and chips, whole genome sequencing (WGS), loop-mediated isothermal amplification (LAMP), and fluorescence in situ hybridization (FISH) methods are the main techniques used for the detection of the antimicrobial-resistant profile [99,104,118].

**PCR-based methods** are among the most efficient and widely used rapid molecular tools to quantify and profile genes encoding resistance in species and genus levels. This method amplifies the target nucleic acid sequence using specific primers that anneal to single-stranded DNA after denaturing the target DNA at a high temperature [129,130]. Advancements in PCR offer a more rapid and robust variation of this technique, such as real-time or quantitative PCR (qPCR), reverse transcriptase PCR (RT-PCR), digital PCR, multiplex PCR (mPCR), and automated PCR. For instance, mPCR offers the advantage of the simultaneous detection of multiple resistant genes through the use of multiple sets of primers [99,118,123,131]. Real-time PCR, or qPCR, allows for the rapid simultaneous detection and quantification of amplified PCR products using fluorescent dyes, eliminating gel electrophoresis [99,132,133]. Automated systems of PCR or qPCR are commercially available and automatically purify the sample, concentrate DNA, and amplify and detect major bacterial genes, confirming antibiotic resistance in less than two hours [99,134].

For two decades, PCR-based techniques have been used as the gold standard for the detection of β-lactam resistant genes in *Enterobacteriaceae*. For example, multiplex PCR was developed to detect 11 acquired genes encoding carbapenemase (*bla_IMP_*, *bla_VIM_*_,_
*bla_NDM_*, *bla_SPM_*, *bla_AIM_*, *bla_DIM_*, *bla_GIM_*, *bla_SIM_ bla_KPC_*_,_
*bla_BIC_*, and *bla_OXA-48_*) using three different multiplex reaction mixtures [108,123,135]. Several automated systems were also developed to identify the target genes [108,123,135]. Real-time multiplex PCR or qPCR systems allow a combination of amplification and detection in a single step, limiting contamination risks. GeneXpert is an automated real-time PCR platform that uses the Carba-R assay and can detect and quantify numerous bacterial species and several carbapenemase genes from rectal samples [123,136]. The Check-Direct assay has a panel of different multiplex real-time PCR kits using several probes, including narrow and broad-spectrum B-lactamase genes [108,123,137]. A broad range of multiplex PCR panels was developed to increase their analytical performance without requiring skilled personnel. However, these PCR-based tests are expensive, limiting their use in low-resource laboratories [123].

**Other molecular methods such as FISH, microarray, WGS, and LAMP** assays have also been used for detecting carbapenem resistance [99,118,138]. FISH is a technique for detecting specific RNA or DNA sequences using dye-labeled oligonucleotide probes visualized by fluorescence microscopy [139]. Microarray-based methods utilize multiple spots on a solid support chip for different oligonucleotides corresponding to resistant genes to detect labeled DNA fragments in a single assay [140]. In the whole genome sequencing (WGS) technique, a whole bacterial sequence is screened for antibiotic-resistant genes and compared with known genes in publicly available databases, allowing the prediction of existing and emerging phenotypic and genotypic resistance [141]. Lastly, the LAMP assay is a simple amplification technique that resolves PCR temperature cycling using a single temperature for target gene amplification. This method produces a large number of DNA copies in a short period [142,143]. LAMP has been used as an alternative to PCR due to its simplicity and cost-effectiveness, especially in low-resource setting laboratories. However, the technique still requires a complex primer design [99,104,138].

**Emerging molecular techniques and automated systems** have been improved to reduce costs and the detection system for β-lactam resistant genes [108,123]. Luminex tech, for example, is a well-established approach based on a colored microsphere-based flow cytometry assay. The method can detect specific alleles, antibodies, or peptides from a single colony [144]. The multiplex oligonucleotide ligation-PCR procedure assists in detecting β-lactam resistant genes and their variations with higher sensitivity and specificity (100% and 99.4%) within 5 h [123]. Further, the LAMP method, using hydroxy naphtol blue dye (LAMP-HNB) and microarray techniques, detects genes encoding carbapenemase with higher specificity and sensitivity at 100% and >90%, respectively [99,108,142,143]. Multiplexed paper-based Bac-PAC is another assay used to categorize the AMR profile of individual strains of CRE by providing a colorimetric readout [145]. The RNA-targeted molecular approach, NucliSENS EasyQKPC test, has also been used for detecting *bla_KPC_* variants within 2 h, at a 93.3% sensitivity and 99% specificity [146]. Another technique, PCR amplification coupled with electrospray ionization mass spectrometry (PCR-ESI-MS), has been used to accurately measure exact molecular masses. With advanced software, the sequence of DNA fragments is reconstructed for accurate identification as well as subtyping of the resistant genes. This technique has been used to identify *bla_KPC_* genes directly from clinical samples in 4–6 h [147]. Lastly, whole genome sequencing methods have been used as the most reliable technique to detect carbapenemase, but the high cost, longer turn-around time, and complex data management limit their use [99,108,148].

Genotypic methods generally offer key advantages, including higher sensitivity and specificity in a short time, increasing their real-world applicability. However, these methods require costly reagents and equipment and need skilled operators [99,103,116,149]. In addition, their sensitivity and selectivity can be affected by specimen debris, resulting in the inhibition of the reaction or false positives [99,123]. Another limitation of many molecular assays is that only known genes can be targeted; phenotypic resistance may be missed by molecular assays, but WGS can help discover novel genes [99,104,116,129]. Further, the limited number of targeted genes is a challenge in molecular tests due to the diversity of carbapenemase-encoding genes. Thus, the target gene is mainly based on the most relevant variant in each geographical area. For example, several commercial kits stated have been developed to detect *bla_KPC_*, the most prevalent carbapenemase in the USA, and may not be used for other genes [123].

### 3.4. Rapid Serological (Immunological) Methods

Immunological assays rely on antibody–antigen reactions to detect bacteria, providing rapid results at a moderate cost [150]. A few methods, including latex agglutination and immunochromatographic assays, have generally been used for the detection of methicillin-resistant *Staphylococcus aureus* (MRSA) [99]. Enzyme-linked immunosorbent assay (ELISA) has also been used to detect either genes or proteins by the combination of the PCR amplification of samples [99,138], which aids in detecting MRSA and CRE [99]. Additionally, a lateral-flow immunochromatographic assay, OXA-48K-SeT, was designed based on the immunological capture of two epitopes that are specific to OXA-48 variants, using colloidal GNPs in 15 min. It has 100% sensitivity and specificity with a detection limit of 10^6^ CFU/mL [150]. In another study, a lateral-flow immunochromatographic assay was used to detect the carbapenem-resistant gene, *bla_OXA-23-like_*, using multiple cross displacement amplification from pure culture as well as clinical sputum samples [151].

Further, a multiplex immunochromatographic test, ICT RESIST-4 O.K.N.V. K-SeT, uses monoclonal antibodies to rapidly detect OXA-48 variants, KPC, NDM, and VIM carbapenemases. This assay has 99.2% sensitivity and 100% specificity from pure culture on Mueller Hilton Agar (MHA) [152]. The immunological assays depend on the level of protein production; accurate results require an enrichment of 18 h [150,152]. However, the diversity of carbapenemase can prevent further developments since initially designed antibodies may not be applicable for targeted antigenic site modification [123]. Although it is a rapid test, it is still costly [123,152], with lower sensitivity and specificity in complex matrices [99,123,138].

### 3.5. Biosensing Techniques

Biosensors, as analytical devices, have emerged as alternative techniques for simple, rapid, cost-effective, and reliable pathogen detection. Biosensors utilize biological or chemical reactions and convert the recognition event into measurable signals for the detection of the target analyte [153,154]. Biosensor types are classified based on their data output system, target analyte, and label dependence [155]. Mainly, biosensors are classified as thermal, mechanical, electrochemical, and optical based on their operating mechanism [118]. Several biosensor applications, particularly electrochemical and optical biosensors, are well documented, but few studies have been developed for antibiotic resistance, especially for carbapenem-resistant bacteria. Biosensor platforms often share the same mechanism, advantages, and disadvantages, and for the purpose of this brief overview, popular electrochemical and optical biosensors are elaborated on. Examples of antibiotic-resistant detection studies are discussed.

**Electrochemical biosensors** utilize the electrical response of bacterial cells; the immobilized bio-recognition element on an electrode interacts with the target, resulting in an electrical signal. Various recognition elements (antibody, phage, aptamer, DNA, etc.), nanoparticles, and signal processing techniques have been used [118,155]. For example, DNA-based biosensors typically use single-stranded DNA immobilized on an electrode with a sequence complementary to the target DNA. The difference between the electrical properties of single-stranded DNA and hybridized double-stranded DNA assists in the detection of the specific target using cyclic voltammetry [156]. Numerous electrochemical biosensors have been used to detect antibiotic resistance. Examples include a study that combined nitrogen-doped graphene with GNPs to detect the human multidrug-resistant gene *MDR1* [157]. In another study, an electrochemical DNA-sensing system identified MRSA based on a MNP/DNA/AuNP hybridization complex using a chronoamperometric signal [158]. Another electrochemical sensor utilized an antibody conjugated with MNPs to detect MRSA from nasal swabs [159]. A label-free electrochemical biosensor detected a PCR amplified *bla_NDM_* gene in carbapenem-resistant *Citrobacter freundii* using impedance spectroscopy [160]. In another study, *bla_KPC_* detection was achieved in *K. pneumoniae* and *E. coli* using voltammetry techniques and sandwich hybridization assays in 45 min at a level of 10^4^ CFU/mL [43].

**Optical biosensors** are widely used platforms for bacterial detection and rely on measuring absorbance, fluorescence, Raman scattering, surface plasmon resonance (SPR), and colorimetry [118]. These biosensors are highly sensitive but can be costly. Surface plasmon resonance (SPR) is the most commonly used assay, which utilizes refractive index measurements due to the excitation of the surface plasmon waves by the interaction of an analyte with its ligand [161]. The technique mainly uses antibodies or DNA as a recognition element. For example, the immobilized single-stranded DNA sequences on the surface bind to their complementary sequence upon hybridization, resulting in a change in plasmon resonance [161]. Raman scattering techniques are also common and measure molecular vibrational, rotational, and low-frequency modes, providing characteristic information about carbohydrates, lipids, proteins, and nucleic acids [118,120,138]. However, they require a higher bacterial concentration and are limited in differentiating closer spectral signals. Recently, the Surface-Enhanced Raman Scattering technique (SERS) has been developed using nanoparticles that enhance Raman signals [120]. This assay can differentiate strains of carbapenem-resistant and susceptible *E.coli* using silver nanoparticles [120], gold nanostars [162], and gold and silver nanorods [163] by comparing their SERS spectral signature with higher specificity and sensitivity. However, these optical biosensors require complex and multivariate data analysis.

The performance of these optical and electrochemical biosensors often depends on the detection limit, sensitivity, specificity, reproducibility, interference response, response time, storage, and operational stability [153,154]. These platforms are highly rapid and often sensitive to their target bacteria. They also reduce or eliminate isolation and culture times, allowing direct measurements from clinical and biological samples [118,138]. However, sensitivity at low-level bacterial loads is still challenging for many biosensor platforms, along with their costly and complex techniques for signal measurements and analysis [118].

**Plasmonic biosensors** that allow colorimetric detection are noteworthy; they offer rapid and simple visual detection within one hour without the necessity of complex and costly equipment [164,165,166]. For example, a study used a plasmonic nanosensor for the colorimetric detection of CP pathogens using gold nanoparticles (GNP) based on carbapenemase activity and pH changes [167]. Here, the GNPs changed color in response to pH and turned to purple, blue, or gray, from red, within 15 min. CRE was detected at a concentration of more than 10^5^ CFU/mL directly from urine and sputum samples within 2.5–3 h. The results were easily distinguished visually and confirmed quantitatively using vis-NIR spectroscopy [167]. Further, DNA-based plasmonic biosensors using GNPs have extensively been used to detect target bacterial DNA. For instance, thiol-capped GNPs were used to detect *Klebsiella pneumoniae* within one hour using an amplified K2A gene [165] and the unamplified DNA of uropathogenic *E. coli* [168]. Dextrin-coated GNPs were used earlier to detect the unamplified DNA of *E. coli* O157:H7 [164], *E.coli* [169], *Salmonella*
*Enteritidis* [170], and *Pseudoperonospora cubensis* [171] within 30 min. Further, dextrin-coated GNPs have recently been used to detect KPC-producing bacteria (~10^3^ CFU/mL) from clinical isolates with 79% sensitivity and 97% specificity [172]. Plasmonic biosensors allow the detection of pathogens and resistant bacteria in a short time without complex and costly equipment requirements. However, further attention is needed to detect the resistant genes from clinical and biological samples to improve their accessibility and applicability [164,172].

## 4. Surveillance Systems for Control of Antimicrobial Resistance

The WHO initiated the Global Antimicrobial Resistance and Surveillance System (GLASS) in 2015 to strengthen knowledge and develop strategies against AMR [173]. The GLASS has supported a standardized approach for collecting, analyzing, and sharing data regarding antimicrobial resistance at global, national, regional, and local levels. The system provides surveillance approaches with epidemiological, clinical, and population-level data. It incorporates data on AMR in humans, antimicrobial medicines, and AMR in the food chain and the environment [173].

The GLASS has partnerships with the WHO AMR Surveillance and Quality Assessment Collaborating Centers Network (WHO AMR Surveillance CC Network) [173]. This network has a strong collaboration with AMR regional networks, such as the Central Asian and European Surveillance of Antimicrobial Resistance (CAESAR), the European Antimicrobial Resistance Surveillance Network (EARS-Net), the Latin American Network for Antimicrobial Resistance Surveillance (Rede Latinoamericana de Vigilancia de la Resistencia a los Antimicrobianos (ReLAVRA)), and the Western Pacific Regional Antimicrobial Consumption Surveillance System (WPRACSS) [173].

The CDC also tracks AMR threats and collects data on human infections, pathogens, and risk factors with domestic and international partners. This allows for strengthening and sharing among networks of the collected data submitted to WHO [56]. AMR tracking systems of the CDC are the National Antimicrobial Resistance and Monitoring Systems for Enteric Bacteria (NARMS) and the Antibiotic Resistance Laboratory Network (AR Lab Network). The NARM was established in 1996 by the CDC, FDA, and the United States Department of Agriculture (USDA) in partnership with the government to track antibiotic resistance in pathogens from humans, retail meats, and food-producing animals [56,174]. The AR Lab Network was established in 2016, which supports lab testing in healthcare, community, food, and the environment (e.g., water and soil) [56]. The Global Antimicrobial Resistance Laboratory and Response Network (Global AR Lab and Response Network) of the CDC was established in 2021 to improve the detection of existing and emerging AMR threats in humans, foods, animals, and the environment globally [56].

The CDC tracks resistant bacteria (*Salmonella*, *Shigella*, *Campylobacter*, *Vibrio*, and *E. coli* O157) in infected patients [56]. The FDA checks retail meats from grocery stores (chicken, ground beef, ground turkey, pork, shrimp, tilapia, and salmon) for *Salmonella*, *E. coli*, *Campylobacter*, *Vibrio*, *Enterococcus*, and *Aeromonas*. The USDA, along with FSIS and Agricultural Research Services (ARS), tracks *Salmonella*, *E.coli*, *Campylobacter*, *Vibrio*, and *Enterococcus* in food animals at slaughter (chickens, turkeys, cattle, swine) [56,174]. The CDC tests these specified bacterial isolates to determine their resistance profile and routinely tests 12 classes of antibiotics depending on the bacterial type. Among the antibiotics, aminoglycosides, penicillin, carbapenems, macrolides, β-lactam combination agents, cephems, and tetracyclines are commonly used for AST tests. Recently, carbapenems have been added to the list to test the AMR profile in *Salmonella*, *Shigella*, *E.coli*, and *Vibrio* [56]. Due to the major threat of CP-CRE infections and the colonization to public health with global economic and security implications, their rapid diagnostic surveillance is of utmost importance [22,56,175].

The clinical laboratory improvement amendments (CLIA) monitor the regulation of laboratory testing and the certification of clinical laboratories by the Center for Medicare and Medicaid Services (CMS) before any diagnostic testing [176]. Here, CDC plays a role in providing analysis and research protocols, developing technical standards and laboratory practice guidelines, monitoring proficiency testing practices, etc. [176]. Clinical laboratories are to rapidly provide reliable laboratory data to healthcare providers and determine the cause of infections to implement appropriate treatments. In clinical laboratories, the identification of these infections has been conducted by culture or culture-independent diagnostic tests (CIDT) using commercial antigen-based or DNA-based methods [177,178]. Culturing methods have been used to obtain isolates forwarded from clinical laboratories to public health laboratories for additional testing, such as resistant profiles, serotyping, and DNA fingerprints [178].

Healthcare and clinical laboratories establish protocols that rapidly notify the health department, healthcare provider, and infection control staff and work with public health departments on the submission of specimens for testing [179]. For example, clinical laboratories screen their resistance profile with ASTs, phenotypic carbapenemase tests (CIM, or CarbaNP test), MALDI-TOF tests for fingerprints, and PCR-tests to detect carbapenemase genes. The isolates can then be sent to public health laboratories to confirm the identity of bacterial species and to perform additional susceptibility and genomic testing to characterize the isolates. In case of unusual resistance, the isolates can further be sent to regional laboratories for additional testing. This allows for detecting existing and emerging types of antibiotic resistance, tracking resistance changes, identifying outbreaks, and generating stronger data to protect against future resistance threats [179].

The surveillance programs also aid in preventing and controlling possible future outbreaks from food and water sources. Thus, foodborne pathogens and their resistant profile, including CP bacteria, are tracked in many countries. However, there is now a need to monitor non-animal products, such as fresh vegetables and fruits. There is also an emerging need to monitor nonpathogen bacteria that can be a reservoir for antibiotic-resistant genes [35,96]. Thus, the rapid identification of CP organisms, regardless of their pathogenicity in biological samples and its implementation of the surveillance program, is essential to prevent and control possible future endemics or pandemics.

## 5. Gaps in Detection Technology

Several phenotypic and genotypic methods and biosensors have been developed to detect pathogens and antimicrobial-resistant bacteria with advancements in automation and nanotechnology. All these techniques have advantages and disadvantages in cost, rapidity, simplicity, reliability, and applicability (Table 2). There have been new emerging phenotypic, genotypic, and biosensor techniques to detect carbapenemases in a simple and cost-effective manner. However, these detection techniques need to be developed further by improving their sensitivity, specificity, and testing on clinical and biological samples to increase their real-world applicability and accessibility. The plasmonic biosensors typically do not require highly trained personnel, as is usually the case with conventional and rapid phenotypic and molecular techniques. This enhances their applicability in low-resource settings for on-site detection [164,172]. Since the estimated material cost of the plasmonic biosensor is as low as ~USD 2 per test [172], while rapid molecular and phenotypic tests range USD 23–150 and USD 2–10, respectively [123]. Such rapid and cost-effective techniques as screening/diagnostic tests should be implemented in clinical and public health and agricultural and food testing laboratories, especially in low-resource laboratories.

In addition to concerns about accessible detection assays, pre-analytical sample processing, including the separation, enrichment, and purification of bacteria from biological and clinical matrices, is often a significant challenge before detection. Therefore, the concentration of bacteria is essential to ensure sufficient numbers of cells for rapid detection [100,180,181]. Current bacterial separation techniques in biological and clinical matrices include physical methods, such as centrifugation and filtration, and chemical and biological methods, such as dielectrophoresis, metal hydroxides, and magnetic nanoparticles, detailed in the literature [180,181,182]. However, separation techniques have not been extensively documented for resistant bacteria. While some examples exist for separating resistant bacteria from pure cultures [159,183,184,185], data are scarce for biological samples. This could be related to the fact that the antibiotic-resistant bacteria profile has more commonly been tested on pure cultures after retrieving and identifying them. Therefore, the rapid extraction of AMR bacteria directly from clinical and biological samples is needed for their rapid detection.

Among the separation techniques, chemical and biological separation processes have become popular because of their speed, simplicity, and cost-effectiveness. Within these, magnetic nanoparticles (MNPs) have commonly been used to rapidly and effectively extract bacteria from food and clinical samples without centrifugation and filtration [186,187]. MNPs draw attention because of their low-cost, stability, benign nature, biocompatibility, and functionalization with recognition moieties [149,180,186,187]. Such as many chemical and biological separation techniques, MNP-bacterial cell adhesion relies on cell surface characteristics, such as surface charge, hydrophobic or hydrophilic interactions, and antibody or lectin binding sites [180,188,189]. Therefore, cell surface characteristics are important in understanding cell adhesion mechanisms for bacterial extraction and detection.

Cell surface characteristics of the resistant bacteria were investigated in several studies. For instance, some studies showed that the biochemical components of antibiotic-resistant cells are different from those of susceptible bacteria. Raman spectrum is one example where bacterial differentiation or detection was achieved using unique fingerprint patterns [190,191]. Further, various studies have shown that alteration in the biosynthesis of cell wall material, membrane components, and cytoplasmic contents can result in changes in cell surface characteristics, including cell morphology and surface charge [192,193,194,195,196]. These cell morphological characteristics are utilized for detecting the bacterial-resistant profile using AST techniques [102,103,104]. In addition, cell surface charge characteristics are mostly utilized for studying cell attachment, bacterial capture, and detection [194,195,197,198]. Research is still ongoing on understanding the cell characteristics of antimicrobial-resistant bacteria to discover new potential factors. The cell surface properties of carbapenem-resistant bacteria, in particular, have not been documented well. The changes in cell surface characteristics can impact their adhesion or attachment to substrate surfaces depending on their interactions with the environment [194]. Therefore, cell surface characteristics need further attention in relation to cell adhesion mechanisms for developing or improving rapid extraction and detection assays.

## 6. Conclusions and Future Perspective

The emergence and spread of carbapenem-resistant bacteria are a global health issue. Even though carbapenems are used in human medicine, several studies showed that carbapenem-resistant bacteria are found in humans, food-producing animals, foods, water sources, etc., due to the rapid dissemination in the complex web, affecting the health of people. There have been continuing efforts to develop rapid and cost-effective detection methods to prevent and control their spread in the community. Current rapid phenotypic and genotypic methods often require pure culture, costly and complex equipment, and skilled personnel. Immunological and conventional biosensor assays offer rapid and cost-effective detection, but they still require complex techniques for signal measurements and analysis. Recently, plasmonic biosensors have shown promise as a cost-effective, rapid, and simple detection technique by eliminating complex and costly equipment. The biosensors need further attention for their increased applicability and accessibility, especially in low-resource settings.

For rapid and cost-effective detection, simple and rapid bacterial separation plays a major role. The current separation techniques on bacteria need further investigation on their effectiveness for resistant bacteria. In addition, cell surface characteristics affect their separation and cell attachment properties. Overall, further studies are needed to enlighten their cell surface characteristics, bacterial attachment, and separation techniques to develop rapid and cost-effective detection assays. These assays can assist as screening or diagnostic tests in low-resource settings.

## Figures and Tables

**Figure 1 microorganisms-11-01491-f001:**
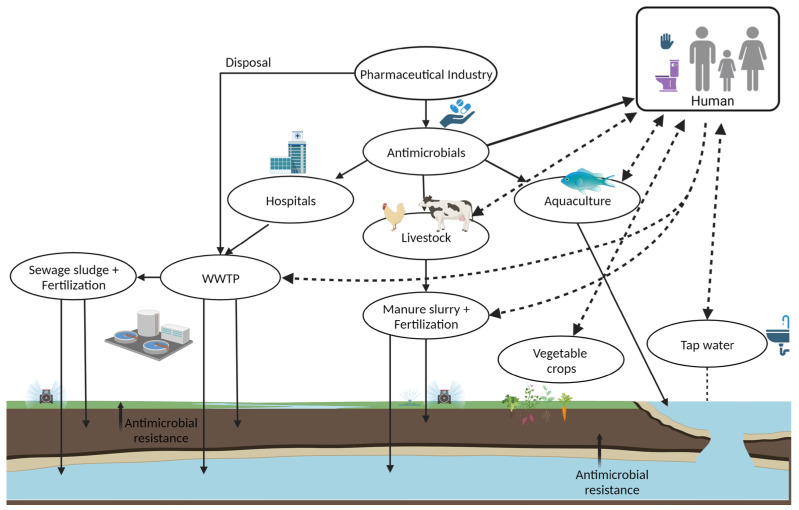
The complex web of the emergence and spread of antimicrobial resistance (created with BioRender.com, accessed on 18 May 2023). WWTP: Wastewater Treatment Plant.

**Table 2 microorganisms-11-01491-t002:** The advantages and limitations of the current and emerging detection techniques for the most common carbapenemases (carbapenem-hydrolyzing enzymes).

Techniques	Advantages	Limitations
**Culture-based methods**	**Simple and cost-effective**	**Time-consuming (>24 h)**
**1.** Improved AST tests: E-test or disk diffusion test [32,107,108]	Detect KPC and MBLs with good sensitivity (>82%) and specificity (>95%)	Insufficient for OXA-48 Require specific reagents and pure culture
**2.** Modified Hodge Test (MHT) * [108,110]	Detects KPC with good sensitivity (>69%) and specificity (>90%)	Insufficient for MBLs Requires pure culture
**3.** Carbapenem-inactivation methods (CIM) * [107,108]	Detect all carbapenemases with higher sensitivity (>90%) and specificity (>95%)	Require pure culture
**4.** Selective media: SUPERCARBA, Colorex KPC, ID Carba, CHROM agar KPC, etc. [112,113,114]	Detect carbapenemases from direct patient samples SUPERCARBA has higher sensitivity (>96.5%)	Variable sensitivity (40–96.5%) and specificity (>50%)
**Rapid phenotypic methods**	**Rapid (<24 h)**	**Costly equipment**
**1.** Colorimetric assay: CarbaNP test and its automated kits * [60,107,108]	Detect carbapenemases with good sensitivity (>70%) and specificity (>80%) Simple, rapid (<2 h), and cost-effective No equipment requirement	Insufficient for OXA-48 Require pure culture
**2.** MALDI-TOF MS * [123,125,126]	Rapidly (1–4 h) detects KPC and MBLs with good sensitivity (>72.5%) and specificity (>95%) Low-measurement cost and simple	Requires data analysis Insufficient for OXA-48 Requires single isolated colonies
**3.** Emerging techniques: BCDA, FC, microfluidic techniques, and Raman spectroscopic techniques [116,119,120,122,123]	Simple and rapid (<4 h) Good sensitivity (>80%) and specificity (>90%) from pure culture	Lower applicability on specimens Insufficient work on carbapenemases
**Genotypic methods**	**Rapid and highly specific (>90%) and sensitive (>90%)**	**Costly and complex equipment**
**1.** PCR-based methods: qPCR, RT-PCR, mPCR, automated PCR (Xpert system, Check-Direct, and Carba-R-assay) [123,131,135] *	Gold standard and rapid (<4 h) Detect and type all carbapenemases directly from specimens	High technical requirements and specific reagents High measurement cost
**2.** Loop-mediated isothermal amplification (LAMP) [123,142]	Simple and moderate cost Applicable in low-resource settings	Specific reagents and complex primer design
**3.** Whole genome sequencing (WGS) [123,141] *	Discovers a new resistance mechanism	Longer turn-around time Complex data management
**4.** Emerging techniques: FISH, microarray techniques, PCR-ESI-MS, and NucliSENS EasyQKPC [116,123,143]	Rapid (<6 h) Detect carbapenemases	Require specific equipment and reagents Insufficient work on carbapenemases
**Immunological Methods** Enzyme-linked immunosorbent assay (ELISA), an Immunochromatographic assay [99,123,138,151]	Rapid and moderate cost Poor sensitivity and specificity directly from specimens	Complex and difficult antibody design due to antigenic site modification
**Biosensors: Emerging Technology**	**Rapid, Simple, and Cost-effective**	**Specific Equipment**
**1.** Electrochemical assays: Impedimetric, potentiometric, and voltammetric [43,156,160] **2.** Optical assays: Raman scattering, SPR, and SERS [118,120,138,161]	Detect carbapenemases Moderate cost	Require equipment for signal processing and data analysis Insufficient work on AMR and carbapenemase detection from pure culture and specimens
2.1. Plasmonic biosensors [167,172]	Rapid, simple, and cost-effective Detect carbapenemases with good sensitivity (78%) and specificity (97%) No equipment requirement	Insufficient work on AMR and carbapenemase detection from pure culture and specimens

* Techniques have been used in diagnostic laboratories (clinical and public health laboratories). AST: antibiotic susceptibility test, MALDI-TOF MS: matrix-assisted laser desorption/ionization time-of-flight mass spectrometry, BCDA: bioluminescence-based detection assays, FC: flow cytometry, FISH: fluorescence in situ hybridization, PCR-ESI-MS: PCR amplification coupled with electrospray ionization mass spectrometry, NucliSENS EasyQKPC: RNA-targeted molecular approach, SPR: surface plasmon resonance; SERS: Surface-Enhanced Raman Scattering technique.

## Data Availability

Not applicable.

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
