# Peer review of "A Review of Carbapenem Resistance in Enterobacterales and Its Detection Techniques"

_microorganisms, 2023, doi:10.3390/microorganisms11061491_

Round 1

Reviewer 1 Report

I suggest if you consider that the high rates of resístance of low incomes and the low probability have PCR in the laboratories of Microbiology

Author Response

Comment : I suggest if you consider that the high rates of resistance of low incomes and the low probability have PCR in the laboratories of Microbiology.

Response: This suggestion is now added in lines 536 (Section on Genotyping Methods) and lines 777-778 (Section on Gaps in Detection Technology). The added line is “low-resource laboratories.”

Reviewer 2 Report

Comments and suggestions for authors

The manuscript entitled "
A Review of Carbapenem Resistance and Its Detection Techniques" comprehensively describes the epidemiological picture for types and antibiotic class carbapenem resistances detected in worldwide.
In my opinion, the manuscript by Oznur Caliskan-Aydogan and Evangelyn C. Alocilja contribute an in-depth description of both phenotypic and genotypic carbapenemase detection techniques already widely found in the literature. The focus of the paper is in an important way on the technological and emerging evolution of carbapenemase detection, namely biosensors.
The technique with biosensors could be of interesting use to circumvent the difficulty in detecting new resistance that phenotypic methods currently may not recognize. All this could negatively impact both diagnostic and epidemiological

In my opinion, in section "5. Gaps in Detection Technology", the authors already highlight where the technique needs to be studied for improvement.
Characteristics such as sensitivity and specificity are of paramount importance in order to identify biosensors as a reliable tool.

Author Response

Comments: The manuscript entitled "A Review of Carbapenem Resistance and Its Detection Techniques" comprehensively describes the epidemiological picture for types and antibiotic class carbapenem resistances detected in worldwide.
In my opinion, the manuscript by Oznur Caliskan-Aydogan and Evangelyn C. Alocilja contribute an in-depth description of both phenotypic and genotypic carbapenemase detection techniques already widely found in the literature. The focus of the paper is in an important way on the technological and emerging evolution of carbapenemase detection, namely biosensors.
The technique with biosensors could be of interesting use to circumvent the difficulty in detecting new resistance that phenotypic methods currently may not recognize. All this could negatively impact both diagnostic and epidemiological
In my opinion, in section "5. Gaps in Detection Technology", the authors already highlight where the technique needs to be studied for improvement.Characteristics such as sensitivity and specificity are of paramount importance in order to identify biosensors as a reliable tool.

Response: Thanks for your comments; we highly appreciate it.

Reviewer 3 Report

General comments of a review manuscript entitled: A Review of Carbapenem Resistance and Its Detection Techniques by Oznur Caliskan-Aydogan and Evangelyn C. Alocilja.

Title is on carbapenem resistance. Perhaps this title can be more specific: A review of carbapenem resistance in Enterobacteriaceae and its detection techniques.

The manuscript emphasises the factors causing the emergence of Carbapenem-resistant Enterobacteriaceae or CPE and its detection methods. The content is thorough to include various industries, environment, agriculture, waste water treatment plan. Two most widely used standard interpretations of AST are CLSI and EUCAST (European Committee on Antimicrobial Susceptibility Testing). Authors are required to include EUCAST in the review manuscript. There is specific section on resistance mechanisms.

Specific comments:

Carbapenem-resistant Enterobacteriaceae is not the same as carbapenemase-producing Enterobacteriaceae. Carbapenemase-producing Enterobacteriaceae or CPE sometimes can have reduced susceptibility to carbapenems, but not considered resistant by the available clinical guidelines for AST interpretation results.

Authors are suggested to use laboratories instead of labs, such in lines 611, 617, 519, 624.

Abstract:

Line 14: Authors need to re-word the sentence. It is due the dissemination of the genes encoding the carbapenemase production.

Carbapenemase is already enzyme. Carbapenemase enzymes are incorrect. Carbapenem-hydrolysing enzymes are correct.  

Introduction:

Great and comprehensive introduction.

The cost burden stated is supporting the reason of this review manuscript.

Line 46: Why it is urgent when there is high presence of antibiotics in the market. Authors are required to provide one to sentence to enable easy to understand the contexts for the readers.

Suggestion for sub-heading of part 2. Part 2. Urgent threat of infections by antimicrobial resistant bacteria: carbapenem-resistant bacteria.

Line 127: Acinetobacter (CRA).

Line 132: human treatment -> perhaps for clinical use in humans.

Line 137: Minor suggestion to include patients in this sentence. -> Carbapenems are typically reserved to use in patients infected with multi-drug resistant (MDR) bacteria…

Line 157: Suggestion for accuracy of the terminology. The carbapenemase “encoding” genes, ……, causing the emergence …...

Line 159: Suggestion for better terminology people -> individuals.

Line 160: surfaces are considered environment.

Line 226-229: The sentences require references.

Line 233: Authors need to check the CP in CP-producing E. coli, …..

CP bacterial was introduced as carbapenemase-producing (CP) bacteria in line 210.

Paragraph of Lines 250-255. Please ensure accuracy of the meaning of the sentences in this paragraph. In the last sentence, it can be simplify to prevent misleading information.

Paragraph of Lines 256-264. Authors are required to improve the grammar to allow accurate contents of this paragraph.

Part 3. Current and emerging detection techniques (Line 265 onwards).

This sub-heading has broad meaning. It doesn’t specify only to CRE. Suggestion for authors to re-word the sub-heading, such as Current and emerging detection techniques of CRE.

Similarly, the first paragraph also needs to be specific towards CRE. The first paragraph on page 7 is an excellent introduction.

Line 279: AST is antimicrobial susceptibility testing.

Great description of Part 3 in page 7 – top of page 14.

Table 2. is good. It will be better if the techniques are supplemented with the use of the techniques at the clinical setting, such as at the diagnostic laboratories. Further details of the total tested isolates, specificity and sensitivity will be beneficial to the readers in trying to choose which techniques to adopt in their clinical settings.  

Complex web was appeared 2x, in the heading of figure 1 and in the conclusions. Authors are required to give a bit of explanation in the manuscript body in paragraph, perhaps around page 3.

There are limited articles from 2023 used as references. Authors are required to check if there are relevant articles from 2023.  

Good quality of English.

Suggestions:

Laboratories should be used instead of labs or lab.

AST = antimicrobial susceptibility testing

Author Response

Thank you for your comments; the authors appreciate the feedback and input. All comments were accepted, and appropriate changes with line numbers are listed below.

General Comments:

Comment 1: General comments of a review manuscript entitled: A Review of Carbapenem Resistance and Its Detection Techniques by Oznur Caliskan-Aydogan and Evangelyn C. Alocilja. Title is on carbapenem resistance. Perhaps this title can be more specific: A review of carbapenem resistance in Enterobacteriaceae and its detection techniques.

Response: Thank you, we have revised the title.

Comment 2: The manuscript emphasises the factors causing the emergence of Carbapenem-resistant Enterobacteriaceae or CPE and its detection methods. The content is thorough to include various industries, environment, agriculture, waste water treatment plan. Two most widely used standard interpretations of AST are CLSI and EUCAST (European Committee on Antimicrobial Susceptibility Testing). Authors are required to include EUCAST in the review manuscript. There is specific section on resistance mechanisms.

Response: CLSI and EUCAST are added in lines 372-376.

Specific comments:

Comment 1: Carbapenem-resistant Enterobacteriaceae is not the same as carbapenemase-producing Enterobacteriaceae. Carbapenemase-producing Enterobacteriaceae or CPE sometimes can have reduced susceptibility to carbapenems, but not considered resistant by the available clinical guidelines for AST interpretation results.

Response: This comment is clarified in lines 174-176 and 186-190.

Comment 2. Authors are suggested to use laboratories instead of labs, such in lines 611, 617, 519, 624.

Response: “Labs” are replaced with “laboratories” in the whole revised manuscript.

Comment 3: Abstract: Line 14: Authors need to re-word the sentence. It is due the dissemination of the genes encoding the carbapenemase production. Carbapenemase is already enzyme. Carbapenemase enzymes are incorrect. Carbapenem-hydrolysing enzymes are correct.  

Response: Thank you, they are corrected in the revised manuscript.

Comment 4: Introduction: Great and comprehensive introduction. The cost burden stated is supporting the reason of this review manuscript.

Response: Thanks, we appreciate your comments.

Comments 5: Line 46: Why it is urgent when there is high presence of antibiotics in the market. Authors are required to provide one to sentence to enable easy to understand the contexts for the readers.

Response: The issue is about unregulated antibiotics in the market. This is addressed in lines 51-53.

Comment 6: Suggestion for sub-heading of part 2. Part 2. Urgent threat of infections by antimicrobial resistant bacteria: carbapenem-resistant bacteria.

Response: Thanks for the suggestion, the sub-heading has been revised.

Comment 7:

Line 127: Acinetobacter (CRA).

Line 132: human treatment -> perhaps for clinical use in humans.

Line 137: Minor suggestion to include patients in this sentence. -> Carbapenems are typically reserved to use in patients infected with multi-drug resistant (MDR) bacteria…

Line 157: Suggestion for accuracy of the terminology. The carbapenemase “encoding” genes, ……, causing the emergence …...

Line 159: Suggestion for better terminology people -> individuals.

Line 160: surfaces are considered environment.

Line 226-229: The sentences require references

Line 233: Authors need to check the CP in CP-producing E. coli, …..

Line 279: AST is antimicrobial susceptibility testing.

Response: All suggestions have been implemented in the corresponding lines on the revised manuscript.

Comment 8: CP bacterial was introduced as carbapenemase-producing (CP) bacteria in line 210.

Response: CP bacteria is now defined as “carbapenemase-producing (CP)” line 190.

Comment 9: Paragraph of Lines 250-255. Please ensure accuracy of the meaning of the sentences in this paragraph. In the last sentence, it can be simplify to prevent misleading information.

Paragraph of Lines 256-264. Authors are required to improve the grammar to allow accurate contents of this paragraph.

Response: We agree these two paragraphs need clarification. They are now edited and combined in lines 324-343.

Comment 10: Part 3. Current and emerging detection techniques (Line 265 onwards). This sub-heading has broad meaning. It doesn’t specify only to CRE. Suggestion for authors to re-word the sub-heading, such as Current and emerging detection techniques of CRE.

Similarly, the first paragraph also needs to be specific towards CRE. The first paragraph on page 7 is an excellent introduction.

Response: The sub-heading title and the first paragraph have been revised as suggested (Lines 344-360).

Comment 11: Great description of Part 3 in page 7 – top of page 14.

Response: Thanks, your comments are highly appreciated.

Comment 12: Table 2. is good. It will be better if the techniques are supplemented with the use of the techniques at the clinical setting, such as at the diagnostic laboratories. Further details of the total tested isolates, specificity and sensitivity will be beneficial to the readers in trying to choose which techniques to adopt in their clinical settings.  

Response: Thanks for the suggestion, the table overall summarizes the detection techniques for carbapenemases with their limitation and advantages. The sensitivity and specificity of techniques are now added into the Table. Also, the techniques used in diagnostic labs are now stated in the Table.

Comments 13: Complex web was appeared 2x, in the heading of figure 1 and in the conclusions. Authors are required to give a bit of explanation in the manuscript body in paragraph, perhaps around page 3.

Response: The paragraph (page 3) is more detailed (Lines 110-123).

Comments 14: There are limited articles from 2023 used as references. Authors are required to check if there are relevant articles from 2023.

Response: The articles from 2023 are added.

Reviewer 4 Report

Caliskan-Aydogan and Alocilja have written a review of carbapenem resistance and basic detection techniques. The topics are important and interesting; such reviews can be found in the literature but since the number of publications in this field is growing very rapidly such review worth publishing. Some modifications should be made, especially for the “carbapenem resistance” part.

1.       Line 9 and entire manuscript - Please replace “antimicrobial-resistant infections” with infections,  caused by resistant bacteria

2.       Line 13 The enzymes cannot be transfer with horizontal gene transfer. Please correct

3.       Line 37 Please replace the “21st century” with “ last decades”

4.       Line 50 The first penicillin-resistant isolate was identified with the  introduction of penicillin so please change “identified” to “increased”

5.       Line 53-60  What have the authors explained – mechanism of action of antibiotics or mechanisms of resistance. The latter seems to be correct, so correct "antibiotics fight bacteria" with "bacteria have different mechanisms to become resistant to antibiotics" or something similar

6.       Line 57 check and correct the third mechanism - “degradation by proteins…”??

7.       Line 72 Please correct - mutant ….. mutation - When a mutation emerges, under selective antibiotic pressure, …..

8.       Line 83 – Please delete “by the formation of sex pili” – this is true for gram-negative but for gram positive pheromones are the important feature for conjugation

9.       Fig1 – what does WWTP mean – please write in abbreviations, Fig1  and section 1.2 – in this figure and section I cannot see the impact of faecal carriage of resistant bacteria. Please include

10.   From section 1 to section 2 the authors should make a subsection (may be as start for the section 2) that explains the importance of beta-lactams , their mechanisms of action, low toxicity and their higher usage and the place of carbapenems in classification of the group. The main mechanisms of resistance to beta-lactams should be also included with explanation for increased ESBL frequency and the need of introduction of carbapenems.  

11.   Line 150-155 the authors can leave the main mechanisms of carbapenem resistance and can include the combination between AmpC and CTX-M enzyme as another mechanism for low level carbapenem resistance.

12.   Line 179 Please don’t specify the exact number of enzymes (write above 75 or 20, however  correct the numbers – for example there are not 22 KPC variants – there are above 75 variants DOI: https://doi.org/10.1128/spectrum.02655-21

13.   Line 179-212 Please completely revised all section for carbapenemases

-          Please include more information for KPC enzymes - the hydrolytic spectrum, their epidemiology (some important lineages, outbreaks) and which are the most common variants. The information for ceftazidime-avibactam action and interesting associations with specific variants can be included.

-           The same should be for NDM, VIM and OXA-48 enzymes. For OXA-48 please include also information for important new variants as OXA-181.

-          Line  201 The first detection of OXA-48 enzyme is in Turkey  in K pneumoniae isolate - Poirel L, Héritier C, Tolün V, Nordmann P. 2004. Emergence of oxacillinase-mediated resistance to imipenem in Klebsiella pneumoniae. Antimicrob Agents Chemother 48:15–22.

-          Line 190 – How “NDM enzymes also may carry other carbapenemase genes”? Please correct

14.   Table 1 please correct the number of enzymes

15.   Line 367-269 Revise the sentence or completely removed.

16.   Line 270 and the whole manuscript “AMR infections” please change to “infections, caused by resistant bacteria”  CRE infections - infections, caused by CRE

17.   Line 279 – please change in  “Antibiotic susceptibility testing”

18.   The gold standard for AST are disk diffusion methods and determination of MIC. Please first explain how they can be used to detect carbapenem resistance, which carbapenem is the most sensitive in detection of carbapenem resistant isolates.

19.   3.1 Section Please reorder the methods – first for specific chromagars, then for MHT and than for combined disk or E test tests. Separate mCIM and explain that this method has better specificity and sensitivity than others.

20.   Line 318-320For chrom agars please check again the specificity and sensitivity, as some authors reported better results for SUPERCARBA medium http://dx.doi.org/10.1016/j.diagmicrobio.2012.10.006

21.   For combined disks please explain the effects of EDTA or DPA and PBA for the carbapenemases group identification

22.   Line 297 and the whole manuscript – Mueller Hinton  is the right name

23.   3.  Section Please separate the different test or techniques explanations, or at least marked their names with bold in the first mention.  

24.   Line 325 Phoenix and Vitec are rapid culture methods – please explain the principle and the duration of the AST for these apparatus. Some new expert software gives new advantages in reporting the results. Please correct the name “BD Phoneix” to “BD Phoenix”

25.   Line 353-354 Please explain how MALDI-Toff can detect the type of carbapenemases and for detection of which group carbapenemases it can be used

26.   Line 389 –“ infectious genes” ?? Please correct

27.   Line 395 The real time PCR  and qPCR  allows for rapid simultaneous. Please include

Author Response

Thank you for your comments; the authors appreciate the feedback and input. All comments were accepted, and appropriate changes with line numbers are listed below.

Caliskan-Aydogan and Alocilja have written a review of carbapenem resistance and basic detection techniques. The topics are important and interesting; such reviews can be found in the literature but since the number of publications in this field is growing very rapidly such review worth publishing. Some modifications should be made, especially for the “carbapenem resistance” part.

Comment 1: Line 9 and entire manuscript - Please replace “antimicrobial-resistant infections” with infections,  caused by resistant bacteria

Response: The term is corrected in the whole revised manuscript.

Comment 2:

  • Line 13 The enzymes cannot be transfer with horizontal gene transfer. Please correct.  
  • Line 37 Please replace the “21stcentury” with “ last decades”
  • Line 50 The first penicillin-resistant isolate was identified with the introduction of penicillin so please change “identified” to “increased”
  • Line 53-60 What have the authors explained – mechanism of action of antibiotics or mechanisms of resistance. The latter seems to be correct, so correct "antibiotics fight bacteria" with "bacteria have different mechanisms to become resistant to antibiotics" or something similar
  • Line 57 check and correct the third mechanism - “degradation by proteins…”??
  • Line 72 Please correct - mutant ….. mutation - When a mutation emerges, under selective antibiotic pressure, …..
  • Line 83 – Please delete “by the formation of sex pili” – this is true for gram-negative but for gram positive pheromones are the important feature for conjugation

Response: These suggestions are all corrected in the corresponding lines on the revised manuscript.

Comment 3: Fig1 – what does WWTP mean – please write in abbreviations, Fig1  and section 1.2 – in this figure and section I cannot see the impact of faecal carriage of resistant bacteria. Please include

Response: Thank you for the suggestion, they have been incorporated in that paragraph (Lines110-123); the meaning of WWPT is added in the figure caption.

Comment 4: From section 1 to section 2 the authors should make a subsection (may be as start for the section 2) that explains the importance of beta-lactams, their mechanisms of action, low toxicity and their higher usage and the place of carbapenems in classification of the group. The main mechanisms of resistance to beta-lactams should be also included with explanation for increased ESBL frequency and the need of introduction of carbapenems.  

Response: This review paper mainly focuses on carbapenems. The types and usage of carbapenems are now added. Overall, this section has been revised for better coherence.

Comment 5: Line 150-155 the authors can leave the main mechanisms of carbapenem resistance and can include the combination between AmpC and CTX-M enzyme as another mechanism for low level carbapenem resistance.

Response: This is now incorporated in that paragraph (lines 182-185).

Comment 6: Line 179 Please don’t specify the exact number of enzymes (write above 75 or 20, however correct the numbers – for example there are not 22 KPC variants – there are above 75 variants DOI: https://doi.org/10.1128/spectrum.02655-21.

Response:  The numbers are corrected in the table and the section text.

Comment 7:   Line 179-212 Please completely revised all section for carbapenemases

- Please include more information for KPC enzymes - the hydrolytic spectrum, their epidemiology (some important lineages, outbreaks) and which are the most common variants. The information for ceftazidime-avibactam action and interesting associations with specific variants can be included.

- The same should be for NDM, VIM and OXA-48 enzymes. For OXA-48 please include also information for important new variants as OXA-181.

- Line  201 The first detection of OXA-48 enzyme is in Turkey  in K pneumoniae isolate - Poirel L, Héritier C, Tolün V, Nordmann P. 2004. Emergence of oxacillinase-mediated resistance to imipenem in Klebsiella pneumoniae. Antimicrob Agents Chemother 48:15–22.

- Line 190 – How “NDM enzymes also may carry other carbapenemase genes”? Please correct

- Table 1 please correct the number of enzymes

Response: This section (2.1) is completely revised; more information on KPC, MBLs, and OXA variants has been added, and the number of variants is corrected in the text and table, as suggested.

Comment 8:    Line 367-269 Revise the sentence or completely removed.

Response: This sentence (267-269) is revised in lines 346-349, and the whole paragraph has been updatd to make it more coherent.

Comment 9:   Line 270 and the whole manuscript “AMR infections” please change to “infections, caused by resistant bacteria”  CRE infections - infections, caused by CRE

Response: This suggestion has been applied in the whole revised manuscript.

Comment 10:   Line 279 – please change in  “Antibiotic susceptibility testing”

Response: It is replaced with antimicrobial susceptibility testing (AST) by another reviewer's suggestion, as seen in line 362.

Comment 11: The gold standard for AST are disk diffusion methods and determination of MIC. Please first explain how they can be used to detect carbapenem resistance, which carbapenem is the most sensitive in detection of carbapenem resistant isolates.

Response: The suggestion is added in lines 368-372.

Comment 12:  3.1 Section Please reorder the methods – first for specific chromagars, then for MHT and than for combined disk or E test tests. Separate mCIM and explain that this method has better specificity and sensitivity than others.

Response: Thanks for the suggestion, the classification was based on general to specific detection of carbapenem resistant profile and then for carbapenemases. AST methods for carbapenem resistance profile, the combined AST test for carbapenemase typing, then other developed specific methods (MHT and CIM), and specific media for the detection of carbapenemases. However, each method in section 3.1 is now clearly specified and revised as suggested.

Comment 13:   Line 318-320For chrom agars please check again the specificity and sensitivity, as some authors reported better results for SUPERCARBA medium http://dx.doi.org/10.1016/j.diagmicrobio.2012.10.006

Response: The sensitivity and specificity of all selective media are added in lines 415-422.

Comment 14: For combined disks please explain the effects of EDTA or DPA and PBA for the carbapenemases group identification.

Response: The comment has been addressed in lines 377-388.

Comment 15: Line 297 and the whole manuscript – Mueller Hinton  is the right name

Response: This comment is addressed in the revised manuscript.

Comment 16:   3.  Section Please separate the different test or techniques explanations, or at least marked their names with bold in the first mention.  

Response: The names of detection techniques are marked boldly in the first mention, as suggested.

Comment 17: Line 325 Phoenix and Vitec are rapid culture methods – please explain the principle and the duration of the AST for these apparatus. Some new expert software gives new advantages in reporting the results. Please correct the name “BD Phoneix” to “BD Phoenix”

Response: Thanks for the suggestion; these techniques are automated AST methods. We want them to categorize in rapid phenotypic tests. However, these methods are now specified as AST rapid-culture methods in the paragraph for clarification (Line 430).

Comment 18: Line 353-354 Please explain how MALDI-Toff can detect the type of carbapenemases and for detection of which group carbapenemases it can be used.

Response: The detection principles and typing carbapenemases have been added in lines 469-481.

Comment 19: Line 389 –“ infectious genes” ?? Please correct

Response: The “infectious genes” is replaced with “genes encoding resistance in species and genus levels”, (Line 509).

Comment 20: Line 395 The real time PCR  and qPCR  allows for rapid simultaneous. Please include

Response:  It is added in line 516.

Round 2

Reviewer 4 Report

The authors answered to some of the comments.

The revision was not marked with different text color.  

There are no association between the specified, in answers to the reviewers, lines and the text.

For comment N3 -  I cannot see where the authors include the explanation for fecal carriage.

Line142-144 I understand that the authors focused on carbapenems, but at least for carbapenems the authors should explain mechanism of action and the structure,

Line 157 should be written “ Carbapenemase-producing”.

Line 172 “carbapenem resistance can be acquired through CTX-M “ please replace the phrase with “carbapenem resistance can be acquired by  combination of CTX-M “

Comment 7 – I also cannot find information for some important lineages of NDM, KPC producers (as ST258, ST512, ST11, ST15 for K pn and ST131 for E. coli)

Author Response

Thank you for your comments; appropriate changes with line numbers are listed below.

Comments:

The authors answered to some of the comments.

The revision was not marked with different text color.  

There are no association between the specified, in answers to the reviewers, lines and the text.

Response: The first revision was made to the manuscript by using the “Track Changes” function, as asked. The uploaded revised file was showing up all changes in red color. With the second round, we were informed that “so sorry that we did not send the reviewer a copy of your manuscript with a revision mark, which we will explain to the Academic Editor after we receive your revised version.”

The second revision was made on the first revised file (from the MDP system), as requested. The second revision part is also done using the “Track Changes’ function. Also, these changes were highlighted in yellow. The revised file is uploaded into the system.

Comment 3: I cannot see where the authors include the explanation for fecal carriage.

Response: That paragraph is reupdated (lines 110-136); the explanation of fecal carriage is included in the paragraph. The figure is also updated.

Comment 4: Line142-144 I understand that the authors focused on carbapenems, but at least for carbapenems the authors should explain mechanism of action and the structure,

Response: The structure of carbapenems and their mode of action were briefly included in lines 161-170.

Comment 5-6:   Line 157 should be written “ Carbapenemase-producing”.

Line 172 “carbapenem resistance can be acquired through CTX-M “ please replace the phrase with “carbapenem resistance can be acquired by  combination of CTX-M “

Response: These suggestions (Line 157 and Line 176) have been implemented in lines 185 and 204, respectively.

Comment 7 – I also cannot find information for some important lineages of NDM, KPC producers (as ST258, ST512, ST11, ST15 for K pn and ST131 for E. coli)

Response: Two brief paragraphs related to the lineages of carbapenemase producers in human pathogens (E. coli and K. pneumoniae sequence types) have been included in Section 2.2, lines 323-340.